# Household food insecurity, maternal nutrition, environmental risks and infants' health outcomes: protocol of the IMPALA birth cohort study in Uganda

Zelalem G Terfa [ID],[1,2] Rebecca Nantanda,[1,3] Maia Lesosky,[1,4] Graham Devereux,[1] Angela Obasi [ID],[5] Kevin Mortimer,[1] Jahangir Khan,[5,6] Jamie Rylance,[1,7] Louis Wihelmus Niessen,[5,8] IMPALA Consortium

ZGT and RN contributed equally. JR and LWN contributed equally.

For numbered affiliations see end of article.

**Correspondence to**
Dr Zelalem G Terfa;
zedgutu@gmail.com; Zelalem.
Terfa@lstmed.ac.uk

## ABSTRACT

**Introduction** In low- and middle-income countries (LMICs), food insecurity and undernutrition disproportionately affect women of reproductive age, infants and young children. The disease burden from undernutrition in these vulnerable sections of societies remains a major concern in LMICs. Biomass fuel use for cooking is also common in LMICs. Empirical evidence from high-income countries indicates that early life nutritional and environmental exposures and their effect on infant lung function are important; however, data from sub-Saharan Africa are scarce.

**Aim** To estimate the association between infant lung function and household food insecurity, energy poverty and maternal dietary diversity.

**Methods and analysis** Pregnant women will be recruited in an existing Health and Demographic Surveillance Site in South-West Uganda. Household food insecurity, sources and uses of energy, economic measures and maternal dietary diversity will be collected during pregnancy and after birth. Primary health outcomes will be infant lung function determined by tidal breath flow and volume analysis at 6–10 weeks of age. Infant weight and length will also be collected.

A household Food Consumption Score and Minimum Dietary Diversity for Women (MDD-W) indicator will be constructed. The involved cost of dietary diversity will be estimated based on MDD-W. The association between household level and mothers' food access indicators and infant lung function will be evaluated using regression models. The Multidimensional Energy Poverty Index (MEPI) will be estimated and used as an indicator of households' environmental exposures. The association between household MEPI and infant lung function will be assessed using econometric models.

**Ethics and dissemination** Ethical approvals have been obtained from Liverpool School of Tropical Medicine (18-059), the Uganda Virus Research Institute Ethics Committee (097/2018) and Uganda National Council for Science and Technology (SS 4846). Study results will be shared with participants, policy-makers, other stakeholders and published in peer-reviewed journals.

## Strengths and limitations of this study

► A key strength is the use of a birth cohort study design that allows prospective assessment of early life exposures and their association with infant lung function.

► We will generate novel data with the potential to inform much needed interventions to address important global health challenges of nutrition and adverse environmental exposures.

► The main limitation is the necessity for clinic-based recruitment of pregnant women for logistic reasons. Women who do not attend antenatal care will be systematically excluded. These women are likely the most disadvantaged and living in extreme poverty, this may represent an important selection bias. The study is located in only one district that is a longstanding surveillance may have resulted in the district being atypical. This could limit the generalisability of our findings.

reduction, food insecurity and undernourishment are still pervasive in sub-Saharan Africa (SSA). The proportion of undernourished people in SSA declined from 28.1% in 2000 to 20.6% in 2010, remained constant until 2015, and then increased to 22.8% in 2018.[1 2] In the SSA region, Eastern Africa has the highest prevalence (30.8%) of undernourishment with more than 133 million affected.[2] The prevalence of total food insecurity is also highest in Eastern Africa, increasing from 58.2% in 2014 to 62.7% in 2018.[2] In Uganda specifically, the proportion of undernourished population declined between 2004 and 2006 and then increased to 39% between 2014 and 2016.[1]

In low- and middle-income countries (LMICs), food insecurity and malnutrition disproportionately affect women of reproductive age, infants and young children.[3–6] Women living in LMICs are particularly at

## INTRODUCTION

In recent decades, despite the world has witnessed significant progress in poverty

BMJ

risk of malnutrition during pregnancy and lactation due to increased nutritional requirements, socioeconomic constraints, societal norms, high intensity of agricultural labour and short interpregnancy intervals.[7]

A child's nutritional status in utero and the postnatal period is inextricably linked to their mother's nutritional status.[8] Intergenerational effects increase infant mortality and have longer-term sequelae.[4 9 10] In Tanzania, about 40% of mothers with under 2-year old didn't consume a diverse diet and 35% of children were stunted.[11] Household food insecurity is a risk factor for iron-deficiency anaemia in infants and toddlers,[12] is associated with small birth size,[13] and adverse infant health outcomes.[14] There is increasing interest in the early life origins and programming of chronic non-communicable diseases,[8] such as the major chronic respiratory diseases of asthma and chronic obstructive pulmonary disease (COPD).[15] In high-income countries, there is a growing body of evidence indicating that suboptimal in utero airway and parenchymal lung development are important risk factors for childhood asthma and probably COPD.[16 17] Maternal nutrition, smoking and exposure to air pollutants have been shown to be associated with reduced infant lung function, changes in neonatal immune and airway epithelial cell function and increased risk of childhood wheezing illness.[18–21]

Poorer households, particularly in low-income countries, tend to have lower-quality diets and select foods of lower nutritional value and lower-quality diets.[22 23] Household-level strategies to cope shortfalls in food availability tend to further reduce both quality and quantity,[24] and food prices can significantly moderate this effect.[25] Food price rises also reduce dietary diversity[26] and consequently nutritional outcomes.[23]

Household air pollution exposure is a predisposing factor for adverse health outcomes in children. In Ghana, prenatal exposure to air pollution is associated with reduced tidal volumes (VTs) and higher minute ventilation (MV) and airway inflammation in 5-week old infants.[27] In Nepal, biomass smoke exposure was associated with lung function deficits in adults.[28] In India, poverty is strongly associated with reduced paediatric lung function, in addition to body mass index, tobacco exposure and biomass use.[29] Household Multidimensional Energy Poverty Index (MEPI), a measure of households deprivation from clean energy source,[30] can be used as a proxy to assess the association between household air pollution and lung health outcomes of young infants. Though varying dimensions of energy poverty indicators have been used in different contexts, studies have shown this parameter is associated with respiratory illness and other health outcome in infants and children.[31–33]

Empirical evidence of the association between early life nutritional and environmental exposures with infant lung function and nutritional outcomes is important to support policy-makers in setting priorities to improve early childhood development and identify intervention mechanisms. Empirical evidence on the effects of in utero nutritional and environmental exposures and infant health outcome are almost exclusively based in higher-income countries.[9 18 19 21 34] In LMICs, few studies have focused on in utero nutritional exposures and non-communicable diseases.[11–13] No studies from SSA have examined the association of households' nutritional and environmental exposure on the respiratory health outcome of infants, particularly infant and maternal lung function. Inequalities in infant lung health outcomes in relation to household socioeconomic status are also unknown in the region. The study presented here will generate a unique body of quantitative evidence in a low-income setting pertaining to the prevalence and distribution of food insecurity, dietary diversity among pregnant/postnatal women, together with cost of dietary diversity (CoDD), energy poverty and its association with infant lung function and nutritional outcomes.

## Study goal and objectives

The study aims to estimate the association of households' food insecurity, mothers' dietary diversity and energy poverty with infant lung function and nutritional outcomes. The specific objectives are as follows:

1. To estimate food insecurity at household level, dietary diversity of pregnant women and related CoDD.
2. To assess the effect of household food insecurity and women's dietary diversity during pregnancy and after birth on infant's lung function and nutritional outcomes.
3. To estimate the association between the household energy poverty of pregnant women and their socioeconomic status with the lung function of infants and mothers.

## METHODS

### Study design

The study will use a prospective cohort design to measure exposures and outcomes among pregnant women and their infants. Data on household food consumption, energy sources, maternal demographics and dietary diversity will be collected during pregnancy and 6–10 weeks after birth. Infant lung function and other health outcome data will also be collected at 6–10 weeks of age. Data collection at 6–10 weeks will enable comparisons with other similar studies[21] and is conveniently the time mothers take their infants to clinics for vaccination.

### Study setting

This study, is part of a larger, multicountry research programme, the International Multidisciplinary Programme to Address Lung Health and TB in Africa (IMPALA).[35] The study will be conducted in the General Population Cohort (GPC), which is a Health and Demographic Surveillance Site (HDSS) in Kyamulibwa subcounty of Kalungu district in South-Western Uganda. The GPC is a population-based open cohort study, established in 1989 by the UK Medical Research Council in collaboration with the Uganda Virus Research Institute to

examine trends in the prevalence and incidence of HIV and its determinants in rural South-Western Uganda.[36] The HDSS comprises 25 rural villages, and a small township of Kyamulibwa Town Council. Annual census and medical surveys have been conducted in this population since 1989.[37]

The main economic activity in the site is agriculture characterised by small farms of bananas, coffee, legumes, vegetables, cassava and potatoes. There is also small-scale cattle, goat and pig farming. A few residents are engaged in small scale trading, selling coffee, food crops and fish. The education level of inhabitants is generally low, with only a third attaining secondary level schooling. There are no special health, economic and social support programmes in the HDSS. The livelihoods and challenges of the residents are generally considered to be typical of rural populations in Uganda.

Five health facilities serve the population with basic medical care, of which three facilities offer maternity services, antenatal care (ANC), deliveries and postnatal care. The HDSS has a pregnancy and birth registration system, whereby pregnant women are registered and followed up until delivery to document birth outcomes. Births are notified to the HDSS administration within 24–48 hours. In Uganda, 97% of pregnant women attend at least one antenatal clinic visit.[38] The existing systems within the HDSS are favourable for conducting cohort studies, including the established birth notification and registration system, periodic censuses and medical surveys. This facilitates the tracking of women through pregnancy, delivery and postpartum, minimising loss to follow-up, and allowing future studies.

### Study population
The study will recruit pregnant women, their infants and households.

### Sampling strategy
This cohort study will be conducted at the three health facilities that offer maternity services in the HDSS. Recruitment will be preceded by a comprehensive programme of community sensitisation. Pregnant women will be identified while attending the antenatal clinics. Consecutive pregnant women will be approached to participate in the study. Written informed consent will be obtained from all participants.

Same day recruitment will be performed if potential participants wish, to minimise inconvenience and costs of repeat travel for participants. However, if potential participants wish to take time to consider participation and/or discuss with members of their family, this will be facilitated.

### Sample size calculation
The sample size estimation is based on estimation of the association between the primary exposures (maternal dietary diversity and household food security) with infant lung function (MV mL/min). The primary outcome will be infant MV, and study power is based on a birth cohort study that related air pollution during pregnancy to infant lung function in which maternal air pollution (PM10) exposure measurements were significantly associated with MV in infants aged 6–10 weeks, and reported a mean (SD) MV of 1401 (242) mL/min.[21]

Maternal dietary diversity is measured by a dichotomous indicator, while household food insecurity is a three-scale categorical variable. The proposed study will have 80% power to detect a 5% difference in minute volume between high and low maternal dietary diversity groups and a 6% difference across the thirds of household food insecurity at the 5% level of significance. With a minimum sample size of 360 (complete data), we will have at least 80% power to estimate moderate effect sizes (OR≥1.5) under a logistic regression model, and >90% power to estimate differences of 5% in continuous infant lung function outcomes with 95% CI, under a linear regression model. We therefore aim to recruit 560 pregnant women to compensate for an anticipated 10% loss to follow-up, a 5% rate of miscarriages, still births and neonatal deaths, and failure to obtain valid lung function test results in 25% of infants.

### Eligibility criteria
#### Pregnant women
All pregnant women, irrespective of gestational age, presenting for ANC services at the selected health facilities and residing within the HDSS will be eligible to participate. Women who will not be available throughout the entire study period (such as those intending to go to another area for delivery of their babies or other reasons) and/or unwilling to participate in the study will be excluded.

#### Infants
All singleton infants will be eligible. It will not be possible to conduct lung function tests for infants with congenital abnormalities of the upper airway for obvious reasons. Therefore, these infants (likely to be very few) will be excluded at analysis of the primary outcome (MV).

### Data collection and analysis methods
The data collection process will follow a number of steps. The data will be collected at four-time points:
- ► Time-point 1: during the antenatal clinic, recruitment point.
- ► Time-point 2: home visit to the pregnant woman households, 1 weeks after recruitment.
- ► Time-point 3: clinic lung function testing for mothers and their infants, 6–10 weeks after birth.
- ► Time-point 4: home visit the women's household during 6–10 weeks after birth.

A summary of participant contacts and required data to be collected at each of the four-time points is presented in flowchart (figure 1). Details of data collection and analytical methods are presented under each of the substudy topics. There will be three substudies as listed below.

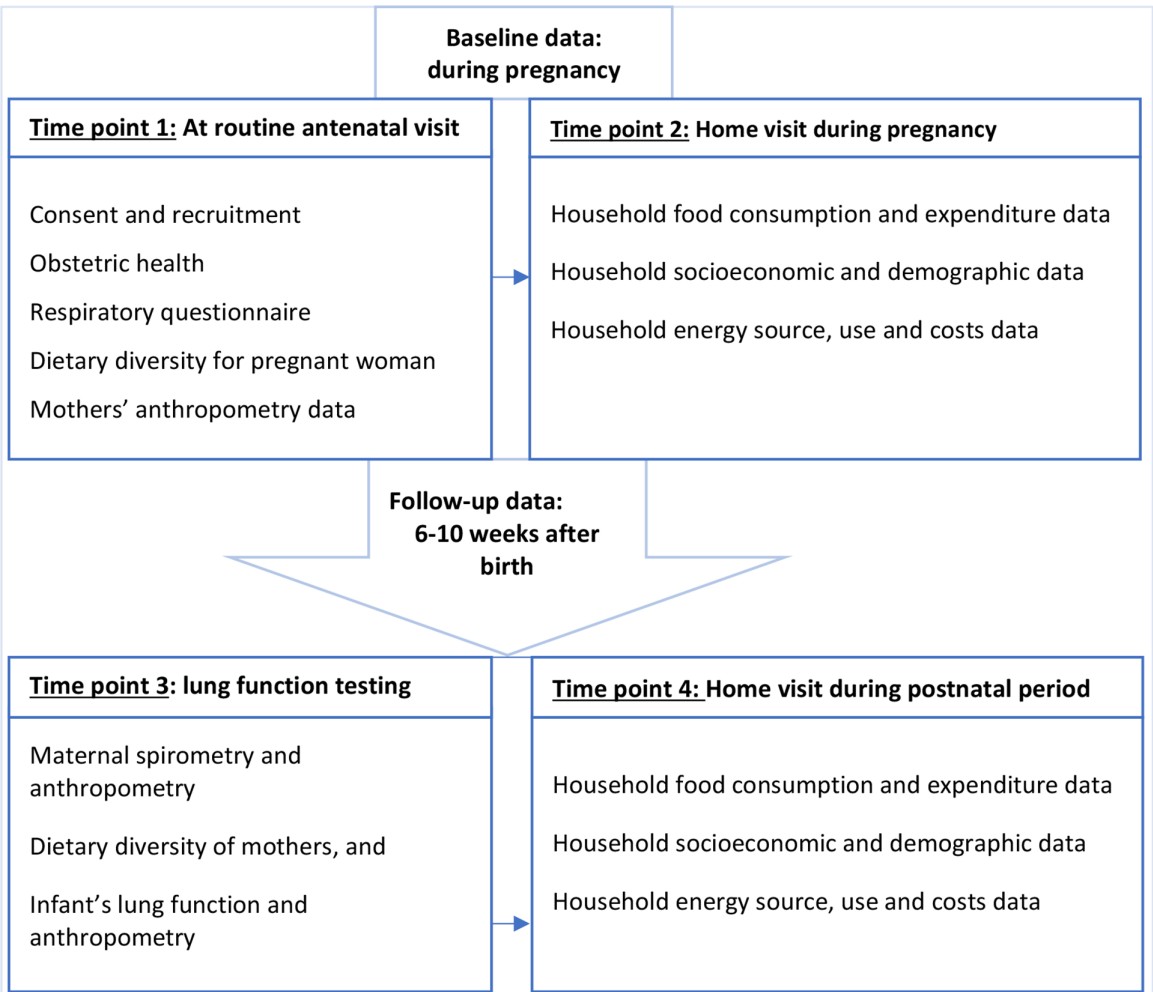

**Figure 1**  Flowchart for summary of data collection process and required data at each of data collection steps.

## Study 1: estimate food insecurity at household level, dietary diversity of pregnant women together with CoDD

The main aim of this substudy is to characterise households' food insecurity and dietary diversity for participating pregnant women. Closely related to dietary diversity of pregnant women, the CoDD will be estimated. Socioeconomic determinants of household food insecurity and women's dietary diversity will also be identified.

### Data collection

Household level food consumption and expenditure data will be collected using the World Bank's living standard questionnaire that has been used in many African countries, including Uganda. Consumption questionnaire will be mapped with Food Consumption Score (FCS) food items and groups. The FCS is a validated tool to assess household level food insecurity. It combines data on dietary diversity and food frequency using 7 days recall data.[39] The main advantage of this tool is that it captures dietary diversity and household food access. It also enables an understanding of the usual consumption behaviour of households.[40 41] FCS will be used to assess household level food insecurity during pregnancy (and after birth). Data on household's food insecurity coping

strategies will also be collected. In addition, separate data will be collected using the Minimum Dietary Diversity for Women (MDD-W) questionnaire, a validated tool,[3] to assess pregnant women's dietary diversity. We will do adaptation work for the MDD-W questionnaire based on local context. Adaptation of the MDD-W questionnaire is helpful for adequate listing of locally available food items, contents of mixed food and food items that are consumed in trivial amounts and to reflect cultural norms, vocabulary and usage of words and phrases that will be easily understood.

Nationally, representative monthly food price data will be obtained from Uganda Bureau of Statistics to estimate the CoDD for women of reproductive age. CoDD is a price index defined around the MDD-W. It is an indicator that provides the least expensive way of meeting the MDD-W.[42]

Anthropometric (weight, height and mid upper arm circumference) and demographic data (including age) will be collected from pregnant women. The same anthropometric measuring equipment will be used across the clinics. As described above, women's anthropometric measurements will be taken at clinics during pregnancy check-up visits. Data on household asset ownership will

be collected. These data will include education, access to land, livestock holdings, type of crops grown, farm and other household assets, non-farm activities of the household. This data will be used to construct household asset index.

## Data analysis

Data will be analysed using both descriptive and multivariable regression models. The household level FCS will be obtained from frequency of consumption of each food group in the household and by assigning standard weight for each food groups. This score will be used to categorise households into three levels of food insecurity status: poor, borderline and acceptable following the standard cut-off points.[39 40 43 44] To construct the MDD-W indicator, food groups and sub-food groups will be aggregated into 10 MDD-W food groups. Woman's dietary diversity will be categorised into two based on standard threshold: scoring≥5 and below the threshold.[3 45 46] The CoDD will be estimated based on rank order optimisation based on food prices within group as described by Masters *et al*.[42] In this technique, the cheapest food items will be selected from each of the 10 food groups then the 5 cheapest food items will be selected. This identifies the cheapest way to achieve MDD-W. This will be done on monthly basis and will be linked with monthly dietary diversity of pregnant women.

Associations of household level food insecurity and women's dietary diversity will be assessed using parametric or non-parametric statistical tests, depending on distribution of the variables. Determinants of household food insecurity, and dietary diversity among women will be assessed using discrete choice based regression models. The household level food insecurity, the dependent variable, is an ordered categorical variable. This suggests the natural choice will be the ordered logit (or probit) regression model. Maternal dietary diversity takes dichotomous variables. Therefore, a multivariable logit (or probit) regression model will be used to identify determinants of dietary diversity in women. Principal component analysis (PCA) will be used to construct households socioeconomic status index, using asset data, to categorise households into socioeconomic quintiles and the method is explained elsewhere.[47 48] Then, distribution of household food insecurity and women's dietary diversity across socioeconomic status will be assessed.

## Study 2: assessing the effect of household food insecurity and women's dietary diversity on infant's lung function and nutritional outcomes

The dynamics of household food insecurity during pregnancy and after birth and its impacts on lung function and the nutritional outcomes measures (birth weight and length) of infants will be assessed. Changes in mothers' dietary diversity during pregnancy and after birth will be thoroughly assessed.

## Data collection

Data collection from study 1 will be repeated in the same individuals 6–10 weeks after birth. In addition, infant

anthropometric measures (weight and length) will be collected. We will also collect data on method of delivery. Preterm babies will also be identified during data collection. Maternal lung function (spirometry) and anthropometric data (weight and height) will also be collected. In addition, infant lung function will be measured. Infant lung function testing will be performed during quiet unsedated breathing using the tidal breath analysis method performed by clinical sciences research team in IMPALA programme. Tidal breathing will be taken as the natural physiological state of undisturbed regular breathing, according to the European Respiratory Society (ERS)/American Thoracic Society (ATS) standards of infant lung function testing.[49] With the sleeping infant is a supine position and neck slightly extended, a face mask will be carefully and gently applied onto his/her mouth and nose. The infant will be given 2–5 min to adapt to the mask before actual measurement of the tidal breaths starts. Tidal breathing will be recorded for a total of 10 min. This time is deemed sufficient to obtain valid information regarding the lung functioning. Tidal breathing measures of MV, VT, respiratory rate (RR) and expiratory flow ratios (tPEF:tE) will be collected using the Exhalyser D with ultrasonic flow metre (Ecomedics, Duernton, Switzerland). The tests will be conducted using the Exhalyser D with ultrasonic flow metre and interpreted using Spiroware V.3.2.

## Data analysis

Household level food insecurity and maternal dietary diversity will be measured using the techniques discussed under study 1. The dynamics in household food insecurity and women's dietary will be analysed using descriptive statistics.

The association between household level and mothers' food access indicators and infant lung function (measured by using MV, VT, RR, tPEF:tE) and nutritional outcomes (weight for age and length for age) of infants will be evaluated using appropriate econometric models. We propose to use multiple linear regression model to estimate the association.

Household socioeconomic status will be categorised based on asset index. PCA will be used to construct household socioeconomic status index. Distributions of lung function and other health outcome across socioeconomic status of households will be examined using a concentration curve.[50 51] The concentration curve gives a more complete picture of socioeconomic inequality in health outcomes. However, it does not give a measure of the magnitude of inequality.[47] Therefore, concentration index, which quantifies the degree of socioeconomic related inequality in a health variable,[50–52] will be used to measure the degree of socioeconomic-related inequality in infant's and mother's lung function and other health outcomes.

These analyses will lead us to identify socioeconomic determinants of infants' lung function and other health outcome (nutritional status of infants).

### Study 3: estimate the association of household energy poverty and their socioeconomic status with lung function of infants and mothers

This substudy aims to assess the association between household energy poverty and the lung function of infants and their mothers. The sources of energy used by households, available/alternative energy sources and households' access to alternative energy sources and implication for risk of respiratory diseases will be explored. Inequalities in infant lung function and nutritional outcomes across socioeconomic status of households will be explored.

### Data collection

Data on household energy source and use will be collected. Attention will be given to energy sources used by households, access to different sources of energy and energy by category of use. We will also collect data on ownership of appliances, entertainment and communication equipment that rely on access to electricity. The lung function of infants will be measured at 6–10 weeks after birth using methods described under substudy 2. The lung function of the mothers will be assessed using Spirometry, performed according to the ERS/ATS standards for Spirometry.[53] Up to eight forced expiratory manoeuvres will be recorded using EasyOn spirometer (ndd, Switzerland), with quality control performed by an experienced clinician. Additional external review will take place for 10% of traces by a respiratory specialist.

### Data analysis

MEPI will be estimated from household energy use and sources data. The MEPI captures and evaluates a set of energy deprivations that affect households. Following Nussbaumer *et al*,[30] the MEPI that is composed of five dimensions representing basic energy services and six indicators of these dimensions will be estimated. Essentially, a household is identified as energy poor if the respective set of deprivation exceeds a predefined threshold. However, there is no standard and agreed threshold. We will categorise households into quintiles based on MEPI and assess the distribution of infants' and women's lung function along the quintiles. Inequalities in nutritional outcome of infants will be assessed using method discussed under substudy 2. The association between energy poverty and infants' lung function will be evaluated using multivariable regression models.

### Patient and public involvement

We appreciate movement towards involvement of patients and the public as coproducers of research. The communities and community leaders were involved in community-level sessions. Patient representatives were not directly involved in the development of this research project. We will seriously consider this in future studies.

### DISCUSSION

The study will provide new and valuable quantitative empirical evidence on the relationships and interactions between household level food insecurity, maternal dietary diversity, energy poverty, and infant lung function and nutritional outcomes in a low-income setting. We are not aware of other similar studies in SSA. This evidence is critical for policy-makers to make informed decisions in identifying and prioritising interventions to address the burden and transgenerational consequences of maternal malnutrition during pregnancy. It will also provide a better understanding of public health implications of energy poverty and indoor air pollution at household level.

A key strength of this study is the use of a birth cohort study design, which allows prospective assessment of early life exposures and their association with infant health outcomes. The necessity for clinic-based recruitment of pregnant women for logistic reasons (availability of lung function testing equipment) will result in women who do not attend ANC being systematically excluded. These women are likely to be the most disadvantaged and living in extreme poverty, this may represent an important selection bias. However, ANC uptake in Uganda is high,[38] which should limit the size of this bias. The fact that our study is located in only one district and that longstanding surveillance[54] may have caused that district to be atypical in other ways, could limit the generalisability of our findings.

However, improving infant and maternal nutrition is a key global health priority. It is also increasingly recognised that poor lung function is associated with increased respiratory and non-respiratory morbidity and mortality across the life course.[54] Our prospective study will generate novel data with the potential to inform much needed interventions to address these important global health challenges.

**Author affiliations**

[1]Department of Clinical Sciences, Liverpool School of Tropical Medicine, Liverpool, UK

[2]Center for Environment and Development, College of Development Studies, Addis Ababa University, Addis Ababa, Ethiopia

[3]Makerere University Lung Institute, Makerere University College of Health Sciences, Kampala, Uganda

[4]Department of Public Health & Family Medicine, University of Cape Town, Rondebosch, South Africa

[5]Department of International Public Health, Liverpool School of Tropical Medicine, Liverpool, UK

[6]School of Public Health and Community Medicine, University of Gothenburg, Goteborg, Västra Götaland, Sweden

[7]Lung Health Group, Malawi-Liverpool-Wellcome Trust Clinical Research Programme, Queen Elizabeth Central Hospital, College of Medicine, Blantyre, Malawi

[8]Department of International Health, Johns Hopkins University Bloomberg School of Public Health, Baltimore, Maryland, USA

**Collaborators** On behalf of The NIHR International Multidisciplinary Programme to Address Lung Health and TB in Africa (IMPALA) Consortium, which in addition to the above named authors comprises: Emmanuel Addo-Yobo, Brian Allwood, Hastings Banda, Imelda Bates, Amsalu Binegdie, Asma El Sony, Adegoke Falade, Bertrand Mbatchou, Hellen Meme, Beatrice Mutayoba, Muhwa Jeremiah Chakaya, Nyanda Elias Ntinginya, S Bertel Squire, Miriam Taegtmeyer, Rachel Tolhurst, William Worodria, Heather Zar, Eliya Zulu, Lindsay Zurba.

**Contributors** ZGT and RN equally contributed to this paper and are joint first authors. LWN and JR are joint senior authors. ZGT, RN, JR, JK, LWN conceptualised

and designed the study. ZGT and RN wrote the protocol and drafted the protocol manuscript with guidance from LWN, JR and JK. GD helped during protocol development and contributed to methodological, dietary and ethical aspects of the study. ML helped with methodological aspect of the study, including sampling design and sample calculations. AO critically reviewed the protocol manuscript and helped during initial stage of designing the study. KM is director of IMPALA and played key role during early conceptualisation of the study. IMPALA consortium members provided useful comments to protocol manuscript.

**Funding** This research was commissioned by the National Institute of Health Research using Official Development Assistance funding (IMPALA grant number 16/136/35). The views expressed in this publication are those of the author(s) and not necessarily those of the NHS, the National Institute for Health Research or the Department of Health.

**Competing interests** None declared.

**Patient and public involvement** Patients and/or the public were not involved in the design, or conduct, or reporting, or dissemination plans of this research.

**Patient consent for publication** Not applicable.

**Provenance and peer review** Not commissioned; externally peer reviewed.

**ORCID iDs**
Zelalem G Terfa http://orcid.org/0000-0001-7932-5841
Angela Obasi http://orcid.org/0000-0001-6801-8889

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
