## [Reviewer comments · BMJ Open]

ARTICLE DETAILS

TITLE (PROVISIONAL)	Household food insecurity, maternal nutrition, environmental risks and infants' health outcomes: protocol of the IMPALA birth cohort study in Uganda
AUTHORS	Terfa, Zelalem G.; Nantanda, Rebecca; Lesosky, Maia; Devereux, Graham; Obasi, Angela; Mortimer, Kevin; Khan, Jahangir; Rylance, Jamie; Niessen, Louis

VERSION 1 – REVIEW

REVIEWER	Park, Yoo Kyoung Kyung Hee University, Medical Nutrition
REVIEW RETURNED	08-May-2021

GENERAL COMMENTS	The purpose of this current protocol manuscript is to estimate the association between household food insecurity and energy poverty, maternal dietary diversity, and the lung function of infants. The rationale is clearly stated. However, a few things need to be addressed in more detail. 1. When acquiring the data, please collect the age of the pregnant women.2. Pregnancy during adolescence poses risk for premature baby.3. High risk pregnancy (who has complications etc) also may cause preterm delivery.4. Most of these preterm babies have compromised lung function. The authors need to describe all of these factors when analyzing the data.4. The method of delivery needs to be monitored also. I am sure that the researchers will collect those variables, but, did not quite describe within the manuscript. Please revise.
--

REVIEWER	Rohloff, Peter Brigham & Women's Hospital and Children's Hospital
REVIEW RETURNED	26-May-2021

GENERAL COMMENTS	This is a nice protocol on an important topic and I enjoyed reading it. I have a few comments that I hope will be helpful: - The intro would be strengthened by including some discussion of known or hypothesized biological mechanisms (fetal organogenesis, growth restriction etc) by which malnutrition and environmental pollution impact fetal/neonatal lung development and function
---

	 - Under outcomes, measures, and sample size calculation, the precise spirometric components that are being used here need to be given, rather than “lung function” - Figure 1 references spirometry in women, but these details/methods are not described in the text - In the descriptions of the statistical methods to be used, the repeated use of the term “econometric” might be confusing to readers of a biomedical journal. If the plan, for example, is to do a multivariable logistic regression, I think that can just be said as such. If econometric implies something subtle that I don’t understand completely then please do explain in more detail as it is not a term I’m accustomed to finding in this type of study . - A perhaps more important question on the proposed statistical methods is where there will be use of random effects models to control for clustering by household/dyad or recruitment location/clinic - Since the infant spirometry is critical here, I think it needs to be described in quite a bit more details, including the machine/apparatus that will be used and the spirometric components that will be measured and analyzed – and see my point above about unclear definition of what the primary outcome “lung function” relates to on the spirometer (a pulmonologist would know based on the numbers in the sample size calculation, but others would not). - To me the biggest weakness of the protocol is the justification that MEPI is tied to lung function. The cited article links to MEPI but not to studies of its association with lung function. I would strengthen this methodologic justification, and especially discuss whether there is preliminary evidence that this proxy measure is validated for Uganda - The biggest limitation of the study is that no actual measurements of air quality/pollution will be made. I think this should be addressed.
--	---

REVIEWER	Imam, Abdulazeez Medical Research Council Unit The Gambia at the London School of Hygiene and Tropical Medicine, Vaccines and Immunity
REVIEW RETURNED	13-Jun-2021

GENERAL COMMENTS	Thank you for inviting me to review this protocol. This is a novel and important piece of research which will address an evidence gap on factors affecting early life lung health and anthropometry in young infants. Both of these have important implications for life course. I have minor comments - Abstract - The Abstract is structured and balanced but is incomplete. There is no mention of general health outcomes which are in your study objectives. Methods - This is well described but can be improved upon.  1.) Researchers plan data collection in pregnancy but have not defined a period for their data collection e.g. Estimated gestational age (EGA) 24 - 28 weeks. There needs to be standardisation of data collected during pregnancy, the same way postnatal data collection was defined as 6 to 10 weeks. Data such as maternal weight would be highly correlated to pregnancy EGA. 2. Authors report ANC attendance at 97%. How complete is the HDSS birth notification as this is integral to the research? 3. The reason for selecting primary outcome assessment at 6 to 10 weeks is not stated. Is this to allow for comparability with other studies?
--

	4. The researchers propose to consent and recruit pregnant women on the same day. In settings such as the one described, consent is typically a communal affair, particularly for young infants. The researchers risk losing multiple children to follow-up after delivery, particularly if fathers or prominent family figures do not buy into the research. It might be prudent to have a modified consenting, where mothers are sensitised at initial visits and provided information leaflets and their permission is taken to contact fathers or any prominent decision-makers for sensitisation. Sensitised mothers might then consent at a subsequent visit. 5. Authors do not describe exclusion criteria for mothers particularly maternal characteristics that can affect lung health. Please see: Gray D, Willemse L, Visagie A, Czövek D, Nduru P, Vanker A, Stein DJ, Koen N, Sly PD, Hantos Z, Hall GL. Determinants of early-life lung function in African infants. Thorax. 2017 May 1;72(5):445-50. Notable among these possible confounders are maternal HIV and smoking. HIV, in general, is of serious concern in Uganda. 6.) Home visits during pregnancy should also ideally be standardised and have a visit interval/ window .i.e time point 1 and time point 2 should have a defined window. 7.) Food insecurity classification into poor, borderline and acceptable is based on? This does not come out clearly 8.) The word general health outcome in objective 2 and elsewhere comes across as too broad to describe just post-natal length and weight. Do authors want to consider a different term such as anthropometry or postnatal anthropometry perhaps? 9.) Please correct error - Pg 9 line 55 - 'height' instead of 'hight'
--	---

REVIEWER	Lufumpa, Nakawala University of Birmingham College of Medical and Dental Sciences, Institute of Applied Health Research
REVIEW RETURNED	16-Jun-2021

GENERAL COMMENTS	This is a both an interesting study and important work. I look forward to reading about the findings of this work. I have attached a document detailing a few major and minor comments to improve the readability of the protocol and ensure that this study is adequately detailed within the context of existing research.
--

VERSION 1 – AUTHOR RESPONSE

Reviewer: 1

[Dr. Yoo Kyoung Park, Kyung Hee University]

We are grateful to the reviewer for taking their time to read the manuscript. We received invaluable comments and feedback from the reviewer. In the following lines, we highlight how we addressed all comments raised by the reviewer.

Note: our responses are written in blue italic.

The purpose of this current protocol manuscript is to estimate the association between household food insecurity and energy poverty, maternal dietary diversity, and the lung function of infants. The rationale is clearly stated. However, a few things need to be addressed in more detail.

Thank you.

1. When acquiring the data, please collect the age of the pregnant women.

Pregnancy during adolescence poses risk for premature baby.

Age of the pregnant women is one of the key variables to be collected in this study, but it

was not explicitly mentioned. We have now indicated that demographic data of pregnant women will be collected.

2. High risk pregnancy (who has complications etc) also may cause preterm delivery.

The data about obstetric history and examination findings will be collected. This will be used to identify high risk pregnancies such as pre-eclampsia, antepartum haemorrhage/bleeding, and multiple pregnancy that are associated with preterm births.

3. Most of these preterm babies have compromised lung function. The authors need to describe all of these factors when analyzing the data.

We appreciate this input from the reviewer. We will identify preterm babies during data collection as they will be at risk of bronchopulmonary dysplasia. We will consider this during data analysis, while being inclusive as possible for all pregnant women recruited. In addition, we will also collect data on gestational age at delivery to account for the evidence from high income countries is that even a week or two of prematurity is associated with reduced infant lung function.

4. The method of delivery needs to be monitored also.

We do collect data on method of delivery. We have now highlighted the data collection plan to collect data on method of delivery in the manuscript.

I am sure that the researchers will collect those variables, but, did not quite describe within the manuscript.

The reviewer is correct. The investigators team is experienced in the recruitment of cohorts of pregnant women and the follow up of their children in order to identify associations between maternal exposures and childhood lung disease. Many established maternal parameters are quantified they had not all been described in the manuscript.

Reviewer: 2

Dr. Peter Rohloff, Brigham & Women's Hospital and Children's Hospital

We are thankful to this reviewer for taking time to read the manuscript and to give us detailed feedback. We believe comments and feedback we received from the reviewer will enhance quality of the manuscript. In the following lines, we try to highlight how we addressed all comments raised by the reviewer.

Note: our responses are written in blue italic.

This is a nice protocol on an important topic and I enjoyed reading it. I have a few comments that I hope will be helpful:

- The intro would be strengthened by including some discussion of known or hypothesized biological mechanisms (fetal organogenesis, growth restriction etc) by which malnutrition and environmental pollution impact fetal/neonatal lung development and function

We have now included in the introduction the importance of sub-optimal in utero lung development as a risk factor for childhood asthma and almost certainly COPD in adults, we have also mentioned the possible effects of maternal nutrition, smoking and air pollution exposure on infant lung function, airway epithelial cell and immune function, and childhood wheezing disease. We have added suitable references.

- Under outcomes, measures, and sample size calculation, the precise spirometric components that are being used here need to be given, rather than "lung function"

We mentioned the specific outcome (minute ventilation) in the second paragraph of the section. We have changed lung function in the first paragraph to minute ventilation and explained that this parameter was associated with maternal PM10 exposure in the cited study.

- Figure 1 references spirometry in women, but these details/methods are not described in the text

Thank you for pointing out this. This is now addressed in the manuscript (under sub-study 3)

- In the descriptions of the statistical methods to be used, the repeated use of the term "econometric" might be confusing to readers of a biomedical journal. If the plan, for example, is to do a multivariable logistic regression, I think that can just be said as such. If

econometric implies something subtle that I don't understand completely then please do explain in more detail as it is not a term I'm accustomed to finding in this type of study . This is very much appreciated. We have now made changes on the manuscript to make it clearer and more specific and use the term 'multivariable logistic regression'.

- A perhaps more important question on the proposed statistical methods is where there will be use of random effects models to control for clustering by household / dyad or recruitment location/clinic

We have no plan to do this as all the recruited women are from separate households and, hence, independent. The local areas are homogeneous as found in earlier analyses and expected differences on attended deliveries among the few participating clinics are minimal. Lung function is measured in one and same clinic. Also, we will construct asset-based SES index to be used as (control) predictor. Not sure what value this RE model will add to the analysis.

- Since the infant spirometry is critical here, I think it needs to be described in quite a bit more details, including the machine/apparatus that will be used and the spirometric components that will be measured and analyzed – and see my point above about unclear definition of what the primary outcome “lung function” relates to on the spirometer (a pulmonologist would know based on the numbers in the sample size calculation, but others would not).

Thank you for this comment. Lung function testing in the infants will be done using Tidal Breath Analysis (TBA) method. Tidal breathing will be taken as the natural physiological state of undisturbed regular breathing, according to the European Respiratory Society (ERS)/American Thoracic Society (ATS) standards of infant lung function testing. With the sleeping infant is a supine position and neck slightly extended, a face mask will be carefully and gently applied onto his/her mouth and nose. The infant will be given 2-5 minutes to adapt to the mask before actual measurement of the tidal breaths starts. Tidal breathing will be recorded for a total of 10 minutes. This time is deemed sufficient to obtain valid information regarding the lung functioning. Tidal breathing measures of Minute ventilation (MV), tidal volume (VT), respiratory rate and expiratory flow ratios will be collected using the Exhalys D with ultrasonic flow metre (Ecomedics, Duernton, Switzerland). These details have been provided in the manuscript.

- To me the biggest weakness of the protocol is the justification that MEPI is tied to lung function. The cited article links to MEPI but not to studies of its association with lung function. I would strengthen this methodologic justification, and especially discuss whether there is preliminary evidence that this proxy measure is validated for Uganda.

This is very important comment, thank you. We agree, the cited articles are not studies directly on MEPI and its association with lung function. The available literature on MEPI and health outcomes is based on various MEPI dimensions and perspectives. Most of available literature is from Europe with focus on fuel poverty (during winter) and studies heat service deprivation. However, our next interest in this research is to assess association between the new MEPI (measure of clean energy source deprivation is defined in sustainability discussions) and infant lung function.

Those papers from Europe (measuring different dimensions of energy poverty and its association with health outcomes) have now been cited in the introduction section.

Given the current global state of knowledge we consider it very opportune, reasonable and scientifically interesting and justified to hypothesise that exposure to clean energy sources will be associated with improved infant lung function. As such, we see the available literature as leading up and part of the scientific context of our study.

It is too early to ask for a validation of MEPI as proxy in relation to health outcomes.

However, given the fact that MEPI measures household's deprivation from clean energy sources, we decided to use it as proxy for households' environmental exposure, to be tested.

- The biggest limitation of the study is that no actual measurements of air quality/pollution

will be made. I think this should be addressed.

Thank you for pointing out this. However, we would like to make it clear that this study protocol is part of large multidisciplinary research consortium. The focus in this present study is different dimensions of poverty and nutritional exposures as determinants of infant lung function. There is also another study that will collect actual data on household (and community) air-quality measurements. Given the state of the research in this domain we excluded this for the present research question and analysis.

Reviewer: 3

Dr. Abdulazeez Imam, Medical Research Council Unit The Gambia at the London School of Hygiene and Tropical Medicine

We are grateful to the reviewer for taking their time to read the manuscript. We received invaluable comments and feedback from the reviewer. In the following lines, we try to highlight how we addressed all comments raised by the reviewer.

Comments to the Author:

Thank you for inviting me to review this protocol. This is a novel and important piece of research which will address an evidence gap on factors affecting early life lung health and anthropometry in young infants. Both of these have important implications for life course.

I have minor comments –

Abstract - The Abstract is structured and balanced but is incomplete. There is no mention of general health outcomes which are in your study objectives.

This is appreciated. We have addressed this.

Methods - This is well described but can be improved upon.

1.) Researchers plan data collection in pregnancy but have not defined a period for their data collection e.g. Estimated gestational age (EGA) 24 - 28 weeks. There needs to be standardisation of data collected during pregnancy, the same way postnatal data collection was defined as 6 to 10 weeks. Data such as maternal weight would be highly correlated to pregnancy EGA.

Pregnant women, regardless of gestational age, will be eligible to participate in the study. Although the 1st trimester is the most critical period during which prenatal insults such as maternal undernutrition greatly impact on the fetus, most women start antenatal clinic attendance in the 2nd trimester. So, for practical reasons we shall enrol pregnant women at any gestational age. In addition, the energy sources and use, as well as eating behaviour/access to food does not change significantly during pregnancy in these rural communities. Therefore, the information on maternal nutrition and energy use collected at any gestation age would be similar.

It is true that maternal weight will be correlated with gestational age, but for purposes of documenting maternal undernutrition, we shall use mid-upper arm circumference, which is the most objective measure of maternal nutritional status.

2. Authors report ANC attendance at 97%. How complete is the HDSS birth notification as this is integral to the research?

The HDDS has a team of community health workers, referred to as village recorders who are involved in birth notification. Each pair of recorders are assigned one village, and every household is assigned a number. With this comprehensive data, the village recorders identify every birth in the assigned households and report to the HDDS administration monthly.

3. The reason for selecting primary outcome assessment at 6 to 10 weeks is not stated. Is this to allow for comparability with other studies?

The time window of 6-10 weeks was chosen in order to be comparable with other studies but also for the very practical reason that this the time when mothers bring their infants to central clinics for vaccination. A sentence to this effect has been added to the manuscript.

4. The researchers propose to consent and recruit pregnant women on the same day. In settings such as the one described, consent is typically a communal affair, particularly for young infants. The researchers risk losing multiple children to follow-up after delivery,

particularly if fathers or prominent family figures do not buy into the research. It might be prudent to have a modified consenting, where mothers are sensitised at initial visits and provided information leaflets and their permission is taken to contact fathers or any prominent decision-makers for sensitisation. Sensitised mothers might then consent at a subsequent visit.

This is very important, thank you. We plan same day consent and recruitment where potential study participants are happy and free to do. However, if potential participants express a wish to take time to think things over and return another day, this will be facilitated as discussed in the protocol. Therefore, potential study participants who feel they need to discuss with their spouse will be allowed to do so and make their decision together. It is also important to remember that there will be four data collection points. The first data collection point will be at clinics, and we plan to collect data about pregnant women only (still they could take time to discuss with their spouse). Separate consent will be obtained from spouses/guardians for data to be collected from the household and for postnatal data collection. Therefore, potential study participants could discuss with their spouses if they wish to.

We also plan to conduct community engagement involving local leaders, village level study facilitators, and wider community to sensitize the study project. From our local knowledge, community engagement plays important role in informing study projects to the community and the community spreads words about the project. We hope this will also help to encourage active participation in the study and to reduce loss to follow-up. In reality, this is working well and there have been no issues.

A sentence stating that community sensitisation will precede recruitment has been added in the methods

5. Authors do not describe exclusion criteria for mothers particularly maternal characteristics that can affect lung health. Please see: Gray D, Willemse L, Visagie A, Czövek D, Nduru P, Vanker A, Stein DJ, Koen N, Sly PD, Hantos Z, Hall GL. Determinants of early-life lung function in African infants. *Thorax*. 2017 May 1;72(5):445-50.

Notable among these possible confounders are maternal HIV and smoking. HIV, in general, is of serious concern in Uganda.

We are aware of this paper (indeed Diane Gray and Heather Zar trained our team). In order to reflect all of the factors that could influence infant lung function in this LMIC setting we have kept exclusion criteria to a minimum. Therefore, HIV and smoking status will not be exclusion criteria in this study. However, we plan to collect data on smoking behaviour of women and their HIV status. These will be used as control variables during analysis.

6.) Home visits during pregnancy should also ideally be standardised and have a visit interval/ window .i.e time point 1 and time point 2 should have a defined window.

Apologies, this was not indicated in the manuscript. Our plan is to complete data collection of time point-2 (home visit during pregnancy) within seven days after recruitment. We have now made changes on the manuscript to reflect this.

7.) Food insecurity classification into poor, borderline and acceptable is based on? This does not come out clearly

The categorization of households will be based to the food consumption score (FCS) that will be calculated based on frequency of consumption of food groups by households during seven days and standard weight for each food groups. The value of FCS also depends on households' oil and/or sugar consumption behaviour. Therefore, classification of food insecurity status will depend on consumption behaviour of households, in addition to frequency and weight of food groups consumed. To avoid unnecessary details in the manuscript, we have now indicated that we will follow the standard approach for food insecurity status categorization. We hope it is now clearer in the manuscript, too.

8.) The word general health outcome in objective 2 and elsewhere comes across as too broad to describe just post-natal length and weight. Do authors want to consider a different

term such as anthropometry or postnatal anthropometry perhaps?
We have used the term nutrition outcome instead of 'general health'.
9.) Please correct error - Pg 9 line 55 - 'height' instead of 'hight'
Thank you. We have corrected it.

Reviewer: 4

Ms. Nakawala Lufumpa, University of Birmingham College of Medical and Dental Sciences
We are grateful to the reviewer for taking their time to read the manuscript. We received invaluable comments and feedback from the reviewer. In the following lines, we try to highlight how we addressed all comments raised by the reviewer.

VERSION 2 – REVIEW

REVIEWER	Rohloff, Peter Brigham & Women's Hospital and Children's Hospital
REVIEW RETURNED	09-Dec-2021
GENERAL COMMENTS	Authors have responded comprehensively
REVIEWER	Imam, Abdulazeez Medical Research Council Unit The Gambia at the London School of Hygiene and Tropical Medicine, Vaccines and Immunity
REVIEW RETURNED	06-Jan-2022
GENERAL COMMENTS	The authors have satisfactorily answered all the queries I raised. Well done.